# Hydrothermal Synthesis of Bismuth Ferrite Hollow Spheres with Enhanced Visible-Light Photocatalytic Activity

**DOI:** 10.3390/molecules28135079

**Published:** 2023-06-29

**Authors:** Thomas Cadenbach, Valeria Sanchez, Daniela Chiquito Ríos, Alexis Debut, Karla Vizuete, Maria J. Benitez

**Affiliations:** 1Colegio de Ciencias e Ingenierías, Universidad San Francisco de Quito, Diego de Robles y Vía Interoceánica, Quito 170901, Ecuador; 2Departamento de Física, Facultad de Ciencias, Escuela Politécnica Nacional, Ladrón de Guevara E11-253, Quito 170525, Ecuador; 3Centro de Nanociencia y Nanotecnología, Universidad de las Fuerzas Armadas ESPE, Av. Gral. Rumiñahui s/n, Sangolquí 171103, Ecuador; apdebut@espe.edu.ec (A.D.); ksvizuete@espe.edu.ec (K.V.)

**Keywords:** bismuth ferrite, hollow spheres, hydrothermal, photocatalysis, Rhodamine B

## Abstract

In recent years, semiconductor hollow spheres have gained much attention due to their unique combination of morphological, chemical, and physico-chemical properties. In this work, we report for the first time the synthesis of BiFeO${}_{3}$ hollow spheres by a facile hydrothermal treatment method. The mechanism of formation of pure phase BiFeO${}_{3}$ hollow spheres is investigated systematically by variation of synthetic parameters such as temperature and time, ratio and amount of precursors, pressure, and calcination procedures. The samples were characterized by X-ray powder diffraction, scanning electron microscopy, energy dispersive X-ray spectroscopy, and UV-vis diffuse reflectance spectroscopy. We observe that the purity and morphology of the synthesized materials are very sensitive to synthesis parameters. In general, the chemically and morphologically very robust hollow spheres have diameters in the range of 200 nm to 2 μm and a wall thickness of 50–200 nm. The synthesized BiFeO${}_{3}$ hollow spheres were applied as catalysts in the photodegradation of the model pollutant Rhodamine B under visible-light irradiation. Notably, the photocatalyst demonstrated exceptionally high removal efficiencies leading to complete degradation of the dye in less than 150 min at neutral pH. The superior efficiencies of the synthesized material are attributed to the unique features of hollow spheres. The active species in the photocatalytic process have been identified by trapping experiments.

## 1. Introduction

In recent years, the development of efficient and sustainable photocatalytic systems has garnered significant attention in the field of materials science and energy research. Harnessing the power of visible light to drive chemical reactions has the potential to revolutionize various applications, such as water splitting, pollutant degradation, and CO${}_{2}$ reduction [1,2,3]. Among the myriad of photocatalysts, semiconductor materials have emerged as promising candidates due to their unique ability to absorb visible light and generate charge carriers with desirable redox potentials. In particular, semiconductor hollow spheres have attracted substantial interest for their distinctive morphology and versatile properties [4,5,6]. In general, due to their intriguing geometry, consisting of a void enclosed within a thin shell, hollow spheres offer a multitude of opportunities for exploration and application across various fields. In the field of nanotechnology, the precisely controlled size, morphology, delimited local voids, and surface chemistry of these structures have enabled applications in areas such as catalysis, drug delivery, energy storage, adsorption, separation, and sensing. Due to their large surface area-to-volume ratio, high catalytic activity, and robustness, facile tuneability and high crystallinity hollow spheres have been used as nanoreactors for confined catalysis in general. In the field of photocatalysis, in particular, the hollow interiors provide efficient light absorption and facilitate the mass transfer of reactants and products, enabling faster-reaction kinetics. Secondly, the increased surface area and optimized crystal facets of hollow spheres offer a higher density of active sites for catalytic reactions, leading to improved overall efficiency. Various wet chemical techniques, such as hard and soft templating, hydrothermal methods, and solvothermal processes, have been employed to fabricate these unique structures [7,8,9,10]. Templating techniques such as the application of pre-fabricated carbon or silica nanospheres require additional multi-step syntheses, which render this procedure not only more time consuming, but also more expensive. Hydrothermal methods in general are characterized by rather mild reaction conditions allowing the precise synthesis of materials with controlled crystal structures, compositions, and morphologies [11]. By careful control of reaction parameters undesirable side reactions or phase transformations are minimized and lead to reduced defects as well as enhanced reaction rate and scaleability. Thus, this method is widely used to synthesize hollow spheres [12,13], one-dimensional materials such as tubes and wires [14,15], two-dimensional materials [16,17], and heterostructured materials [18,19].

The success of semiconductor hollow spheres as visible-light photocatalysts relies on the careful selection of suitable semiconducting materials. In this context, bismuth ferrite (BiFeO${}_{3}$) is a multiferroic semiconductor with a distorted rhombohedral perovskite structure of the space group R3c [20] that has been intensively investigated due to its unique features, such as ferroelectricity, piezoelectricity, and nonlinear optical properties [21,22,23,24]. Furthermore, BiFeO${}_{3}$ has found applications in electronic devices, such as data storage media, multi-state memories, and quantum electromagnets [22,25].

Due to its chemical robustness and narrow band gap in the visible-light region (2.0–2.8 eV) as well as the rather slow electron-hole recombination rate, it has been used in various photocatalytic transformation and degradation reactions [23,26,27,28]. The size and morphology of BiFeO${}_{3}$ materials have a substantial impact on the photocatalytic activity, in which smaller BiFeO${}_{3}$ particles are characterized by higher surface area and often lower band gaps. For instance, the size of the crystal significantly affects the photocatalytic activity due to higher surface area of smaller particles as well as an easy transfer of charges on the surface of the photocatalyst [29]. Huang et al. [30] synthesized BiFeO${}_{3}$ particles with different morphologies, showing that the highest photocatalytic degradation efficiency is obtained for a honeycomb-like morphology. However, due to oxygen defects and surface constraints, smaller BiFeO${}_{3}$ particles are not always the most photocatalytically active ones. In addition, the synthesis of pure single-phase BiFeO${}_{3}$ in any shape and size is still very challenging, mainly due to the volatilization of bismuth ions; thus, secondary phases such as Bi${}_{2}$Fe${}_{4}$O${}_{9}$ and Bi${}_{2}$O${}_{3}$ often appear [31,32]. Consequently, the synthesis of BiFeO${}_{3}$ demands precise control in order to obtain a material with the desired size, shape, purity, and structure to maximize its photocatalytic performance. Combining the unique features of BiFeO${}_{3}$ semiconductor photocatalysts and hollow materials, BiFeO${}_{3}$ hollow spheres with sphere walls in the nanometer scale are expected to have enhanced photocatalytic properties. Due to the above-mentioned challenges in the formation of BiFeO${}_{3}$, the wet chemical synthesis of BiFeO${}_{3}$ hollow spheres is widely underdeveloped. Recently, Qu and co-workers reported the synthesis of BiFeO${}_{3}$ and BiFeO${}_{3}$/Bi${}_{2}$Fe${}_{4}$O${}_{9}$ hollow spheres by a hard-template method using sacrificial carbon nanospheres [33]. The synthesized hollow structures were characterized by increased light absorption and high photocatalytical activity in the degradation of o-chlorophenol. Furthermore, Khan et al. reported the synthesis of hollow BiFeO${}_{3}$ spheres/porous g-C${}_{3}$N${}_{4}$ as photocatalysts for CO${}_{2}$ conversion and Alizarin Red S degradation [34]. In this research, carbon nanospheres were also used as the structure-directing agent. The synthesized spheres are higher catalytically active than corresponding BiFeO${}_{3}$ nanoparticles. The formation of Bi${}_{2}$O${}_{3}$/BiFeO${}_{3}$ composite hollow spheres by annealing Bi@BiFe-glycolate precursors was reported by Chen and co-workers [35]. The obtained hollow spheres exhibited enhanced photocatalytic activity and stability under visible-light irradiation due to their special morphology. The present work describes for the first time the synthesis of BiFeO${}_{3}$ hollow spheres by a facile hydrothermal method. The mechanism of formation of BiFeO${}_{3}$ hollow spheres is systematically investigated by varying synthetic conditions, such as hydrothermal treatment temperature and time, ratio of precursors, pressure, and calcination procedures. The samples were characterized by X-ray powder diffraction (XRD), scanning electron microscopy (SEM), energy dispersive X-ray spectroscopy (EDX), and UV-Vis diffuse reflectance spectroscopy. Finally, the synthesized BiFeO${}_{3}$ hollow spheres were applied in photocatalytic degradation reactions using Rhodamine B under visible-light irradiation.

## 2. Results and Discussion

The optimal parameters to synthesize BiFeO${}_{3}$ hollow spheres were found by systematically studying the influence of different synthesis conditions, as specified in the methodology. In the first series of reactions, the impact of five different hydrothermal treatment times was tested. These samples were synthesized using 2 eq. of citric acid in a 250 mL Teflon-lined stainless steel autoclave. The temperature during the hydrothermal process was kept at 140 °C.

The presence of BiFeO${}_{3}$ (space group R3c, JCPDS card No. 86–1518), Bi${}_{2}$Fe${}_{4}$O${}_{9}$ ( JCPDS card No. 72-1832), and Bi${}_{2}$O${}_{3}$ (JCPDS card No. 27-0050), when varying the treatment time is shown in the X-ray diffractograms (see in Figure 1a). For the shortest hydrothermal treatment time, the only impurity observed is Bi${}_{2}$O${}_{3}$. As the heat-treatment time increases, the amount of Bi${}_{2}$O${}_{3}$ also increases. In addition, a third phase, i.e., Bi${}_{2}$Fe${}_{4}$O${}_{9}$ becomes evident. Therefore, heating the autoclave for 6 h stabilizes the BiFeO${}_{3}$ phase and prevents the formation of Bi${}_{2}$Fe${}_{4}$O${}_{9}$.

Next, the impact of the ratio between the amount of precursors (bismuth nitrate and iron nitrate) and the amount of citric acid was studied. The amount of citric acid was varied while keeping all other factors constant (12 h@140 °C in a 250 mL autoclave). Here, 6 mmol of nitrates and 3 mmol of citric acid were used for the 1 eq sample, while 6 mmol of nitrates and 6 mmol of citric acid were used for the 2 eq sample. The X-ray diffractograms show the purity of the obtained samples, in Figure 1b. As shown in Figure 1b, the samples synthesized by using 1 eq of citric acid contain primarily BiFeO${}_{3}$ with Bi${}_{2}$O${}_{3}$ as the only impurity. When the amount of citric acid increases, Bi${}_{2}$Fe${}_{4}$O${}_{9}$ starts to form during the synthesis in addition to Bi${}_{2}$O${}_{3}$. Furthermore, the sample using 1 eq of citric acid showed a higher crystallinity as noted in the characteristic double Bragg peak of BiFeO${}_{3}$, i.e., (104) and (110), at around 2θ=32°, as well as an improved signal to noise ratio [36]. Thus, for all further experiments, 1 equivalent of citric acid was employed during the synthesis.

In addition, the effect of temperature during the hydrothermal treatment on the purity of the samples was investigated. Figure 2a shows the X-ray diffractograms of the samples for different temperatures (1 eq of citric acid and 12 h of hydrothermal treatment). The sample heated at 140 °C exhibits a higher crystallinity as indicated by narrower diffraction peaks and a better definition of the (104) and (110) double peak at around 2θ=32°. In addition, a slight increase of impurities when increasing temperature is observed. Thus, the temperature of 140 °C was used for all further reactions.

Furthermore, the pressure during the hydrothermal process was evaluated. This is a critical parameter during the synthesis which can be controlled indirectly. The pressure depends on variables such as the amount of solution, the size of the autoclave, and the temperature during the hydrothermal process. Figure 2b shows the X-ray diffractograms of the samples synthesized in autoclaves of different sizes using 1 eq of citric acid and 6 h@140 °C. It should be noted that the quantities of precursors were adjusted according to the size of the autoclave. In both cases, only Bi${}_{2}$O${}_{3}$ impurity was observed. The semi-quantitative analysis with the DIFRACC.EVA software estimates 14.4% of impurities for the sample synthesized in the 50 mL autoclave. The amount of impurities is considerably reduced to 8.1% when using a 250 mL autoclave. This indicates that in the latter case, the lower pressure facilitates the diffusion between the particles by reducing the viscosity of the solution, which promotes solubility and affects crystal growth.

Finally, the influence of different calcination conditions was analyzed. The X-ray diffractograms of the samples calcined with different calcination paths are shown in Figure 3. All samples were calcined with an intermediate temperature plateau at 200 °C for 2 h in order to ensure complete drying of the powder and in order to avoid unwanted combustion reactions [37]. As indicated by a better signal-to-noise ratio and by the better resolution of (104) and (110) peaks, the sample calcined for 1 h at 500 °C with a fast heat rate of 4 °C per minute is preferred due to it higher crystallinity.

To summarize, optimal reaction conditions in terms of phase purity are the usage of 1 eq of citric acid, hydrothermal treatment of 6 h at 140 °C in a 250 mL autoclave with final calcination of 500 °C for 1 h with a fast heat rate of 4 °C per minute and an intermediate plateau at 200 °C for 2 h. The photocatalytic experiments were performed using the catalyst obtained by these conditions (see below).

The scanning electron microscopy images of the samples synthesized, based on different hydrothermal treatment times, are shown in Figure 4. For all treatment times, hollow spheres between 200 nm and 2 μm in diameter are observed. Interestingly, the hollow spheres resulting from a shorter treatment time are made of smaller crystallites, which can be identified on the surface of the spheres. As the hydrothermal treatment time increases, the crystals on the surface of the spheres start to grow, and consequently, the surface of the spheres becomes rougher and increasingly porous. Furthermore, with an increase in treatment time, the spheres tend to grow and merge into each other, forming larger and more irregularly shaped structures which can be explained by an Ostwald ripening process [4,38]. The hollow nature of the synthesized spheres is verified by the presence of incomplete, and thus, open spheres, shown in Figure 4. The wall thickness of the hollow spheres increases from around 50 nm for the samples synthesized by a hydrothermal treatment time of 6 h to approximately 150–200 nm for the samples subjected to a 24 h treatment period. It should be noted that one of the most accepted mechanisms of the formation of mixed metal hollow spheres is described by the nanoscale Kirkendall effect [39].

The elemental composition of the hollow spheres was determined by EDS measurements, and the results are shown in Figure 5 and Appendix A. The energy-dispersive spectra of the hollow spheres show a uniform distribution of Bi, Fe, and O with an approximate ratio of 1:1:3.

As mentioned above, one of the general obstacles in the synthesis of BiFeO${}_{3}$-based materials is to obtain a pure phase material with the desired morphology. In order to purify the synthesized BiFeO${}_{3}$ hollow spheres, the samples were washed with 2M acetic acid as well as 2M nitric acid. After vigorous stirring for approximately 12 min, the precipitate was collected by means of centrifugation and washed several times with de-ionized water. The effect of the performed washing on the purity of the sample is shown in Figure 6. As shown in Figure 6, the quantities of both unwanted phases, i.e., Bi${}_{2}$Fe${}_{4}$O${}_{9}$ and Bi${}_{2}$O${}_{3}$, are significantly reduced. As reported by various other research groups, remaining Bi${}_{2}$Fe${}_{4}$O${}_{9}$ and Bi${}_{2}$O${}_{3}$ impurities can be completely removed by additional acid washing in order to obtain phase pure BiFeO${}_{3}$. In the present case, when Bi${}_{2}$O${}_{3}$ is the only secondary phase, phase pure BiFeO${}_{3}$ is easily obtained by washing the sample with acetic acid (see Figure 6b).

Interestingly, the acid treatment has no effect on the morphology of the material, as shown in Figure 7. When comparing the SEM images before and after the washing procedure, it can be seen that there is no variation in the shape, size, and surface characteristics of the spheres. This clearly underlines the extraordinary robustness of the synthesized BiFeO${}_{3}$ hollow spheres.

The optical properties of the synthesized BiFeO${}_{3}$ were investigated by diffuse reflectance UV-Vis spectra which were then transformed for analysis with the Kubelka–Munk method to obtain the band gap energy. The absorption band edge appears at 555–565 nm, which indicates that the synthesized samples are capable of absorbing considerable amounts of visible light. By application of the tangent line in the plot of the square root of the Kubelka–Munk function vs. photon energy (Tauc plot), the band gap energy of the hollow spheres was obtained (see Figure 8).

The determined band gap energy of 2.22 eV is considerably smaller, and thus, redshifted when compared to bulk BiFeO${}_{3}$. As we discussed in detail previously, a meaningful comparison of band gap values of different BiFeO${}_{3}$ samples is not always possible as morphology, size, phase purity, as well as oxygen vacancies have a strong impact on the band gap [36,40]. The redshift is often a result of the strained nature of nanosized materials as well as symmetry breaking in high-surface-area materials.

Photocatalytic activity

The photocatalytic activity of the synthesized BiFeO${}_{3}$ hollow spheres was evaluated in photocatalytic degradation reactions using Rhodamine B as a model organic pollutant using visible light (427 nm, 440 nm). A typical reaction setup is depicted in Figure 9.

One of the important aspects of the operation of the experiment is that the sample is stirred vigorously during the complete reaction period. The degradation efficiency is evaluated by the ratio C/C0, leading to a normalized initial concentration ratio of 1. It should be noted that in the absence of a photocatalyst the chosen dye, i.e., Rhodamine B, is extremely stable under visible-light irradiation. Thus, no degradation of the dye is observed after irradiating it for a total of 4 h (see Figure 10). Next, the adsorption capability of the synthesized catalyst was evaluated by stirring the reaction mixture composed of the dye and the BiFeO${}_{3}$ spheres in darkness for a total of 5 h. As can be seen in Figure 10, an adsorption/desorption equilibrium was established during the first hour with a total adsorption of approximately 14%. By using a 2-methoxyethanol/H${}_{2}$O mixture, Rhodamine B can be easily desorbed undegraded from the catalysts, which confirms that the initial removal of the dye is due to adsorption [36]. When using visible light as the irradiation source, Rhodamine B is completely removed from the solution in approximately 150 min. It should be noted that after washing the isolated photocatalyst with a 2-methoxyethanol/H${}_{2}$O mixture, no Rhodamine B could be detected confirming the complete degradation of the dye during the photocatalytic reaction. Furthermore, the complete degradation of the organic pollutant in less than 3 h under pH-neutral reaction conditions represents one of the highest photocatalytic activities of unsupported BiFeO${}_{3}$ materials reported to date. Notably, the photocatalytic performances of the BiFeO${}_{3}$ photocatalysts obtained before and after acid washings are nearly identical (see Appendix A). The photocatalytic activity of the synthesized hollow spheres is much enhanced compared to bulk material and BiFeO${}_{3}$ materials free of surface defects [23]. Furthermore, unsupported BiFeO${}_{3}$ nanoparticles with a large surface area and an average diameter of approximately 5.5 nm were shown to have a degradation efficiency of approximately 88% during the same time period [40]. In fact, as previously reported that the photocatalytic activity of BiFeO${}_{3}$ actually decreases with a reduction in particle size as a result of crystal defects and local distortions [36,40,41]. The much-superior photocatalytic activity can be explained by the hollow nature of the photocatalyst. It has been shown that these hollow spheres in general show improved mass transfer rates, and thus, facilitate the transport of reactants and products to and from the catalyst surface [4,5]. In photocatalytic applications, hollow spheres have been shown to improve light scattering, and thus, absorb more light. In addition, the internal cavity is known to promote further scattering and internal reflection of light, leading to effective light trapping within the sphere and consequently to a higher light-harvesting capability compared to nonhollow analogs. Furthermore, the thin walls of the synthesized hollow spheres allow shorter charge carrier lengths, which decrease the electron-hole recombination before excited electrons reach the catalyst surface [4].

In order to analyze the effects of the catalyst concentration on the degradation reactions, we performed a series of photocatalytic experiments varying the concentration of the catalyst from 0 to 3 g/L. Figure 11 shows the dependence of the catalyst concentration on the degradation of Rhodamine B after 2 h. For low concentrations, the removal percentages significantly decreased due to the lack of active sites that trigger the photocatalytic reactions. As the concentration increases, a region of high photocatalytic efficiency (1 g/L–1.75 g/L) is reached peaking at a catalyst concentration of 1.5 g/L. When the concentration is increased further, the removal efficiency decreases as a result of an increased opacity of the suspension, which is in agreement with previously reported findings. Thus, the optimal concentration of BiFeO${}_{3}$ hollow spheres as photocatalyst for the degradation of Rhodamine B is 1.5 g/L, leading to complete degradation of the organic pollutant in 2 h.

The reusability and the stability of the synthesized BiFeO${}_{3}$ hollow spheres were evaluated in the photocatalytic degradation reactions of Rhodamine B (Figure 12). Here, after a completed photodegradation reaction, the catalyst was separated from the suspension by centrifugation, and washed repeatedly with water and ethanol. After drying the obtained powder, it was reused in the next catalytic experiment using freshly prepared Rhodamine B solution. This process was repeated for a total of four cycles. As shown in Figure 12, the photocatalytic efficiency remains unchanged in all experiments, which demonstrates the reusability and overall stability of the synthesized BiFeO${}_{3}$ hollow spheres.

In water-based semiconductor photocatalysis, excited electrons can react with dissolved oxygen to superoxide anion radicals (• O${}_{2}^{-}$) and hydrogen peroxide H${}_{2}$O${}_{2}$, whereas the formed electron-hole (h+) can continue to react with H${}_{2}$O to hydroxyl radicals (• OH) [1,2]. Furthermore, the hydroxyl radical can also be formed by the disproportionation of the superoxide anion radical; while the hydroxyl radical was considered to be the main active species, in-depth mechanistic studies have shown that all of the above species can participate in the degradation of organic molecules. To shed light on the mechanism and active species involved in the present photocatalytic degradation reaction, we performed standard trapping experiments with different scavengers. The addition of scavengers, such as the hydroxyl radical (• OH) scavenger tert-butyl alcohol (TBA, 2 mM), the superoxide radical scavenger benzoquinone (BQ, 0.5 mM), and the hole scavenger ethylene diamine tetraacetic acid (EDTA, 2 mM), resulted in all cases in significant decreases of the dye removal efficiency, as shown in Figure 13. In the case of EDTA addition, only approximately 20% of Rhodamine B was removed, whereas the usage of BQ and TBA lead to a Rhodamine B removal of 45% and 73%, respectively. These findings confirm that all three species, i.e., photogenerated holes (h+), hydroxyl radicals (• OH), and superoxide radicals (• O${}_{2}^{-}$), are catalytically active species in the present system. Furthermore, the photocatalytic degradation of Rhodamine B was significantly enhanced by the addition of the electron scavenger AgNO${}_{3}$. In the presence of AgNO${}_{3}$, 95% of Rhodamine B was removed after 90 min and total degradation was observed after 120 min. This enhanced degradation activity is explained by an improved electron-hole separation due to the consumption of excited electrons [42]. A schematic representation of the mechanism is shown in Appendix A.

Additionally, in all cases, the formation of intermediate species as a result of the cleavage of ethyl groups and the removal of the conjugated chromophore is confirmed by a blueshift of the maximum peak intensity for Rhodamine B from 553 nm to 548 nm.

## 3. Materials and Methods

### 3.1. Synthesis of BiFeO${}_{3}$ Hollow Spheres

In a typical synthesis, bismuth nitrate pentahydrate Bi(NO${}_{3}$)${}_{3}$(H${}_{2}$O)${}_{5}$ of purity ≥98% (Sigma-Aldrich, St. Louis, MO, USA molecular weight = 485.07 g/mol), iron nitrate nonahydrate Fe(NO${}_{3}$)${}_{3}$(H${}_{2}$O)${}_{9}$ of purity ≥99.95% (Sigma-Aldrich, molecular weight = 404.00 g/mol) and citric acid of purity ≥99.5% (Sigma-Aldrich, molecular weight 192.124 g/mol) were dissolved in ethylene glycol C${}_{2}$H${}_{6}$O${}_{2}$ of purity 99.8% (Sigma-Aldrich). After stirring the solution at 250 rpm for 20 min, urea CO(NH${}_{2}$)${}_{2}$ (molecular weight = 60.06 g/mol) and the necessary amount of distilled water was added to form a solution (1:19) of ethylene glycol and water. Following the order of dissolution of the precursors guarantees a homogeneous solution before the hydrothermal treatment. After stirring the solution for 20 min, the sample was transferred into an autoclave and heated accordingly. The autoclave was allowed to cool down naturally to room temperature and the sample was centrifuged at 1500 rpm for 8 min. After isolating the obtained powder, the sample was washed with a water/ethanol mixture (19:1). This process was repeated six times. Then, the precipitate was collected and dried in a ventilated oven at 80 ${}^{{}^{\circ}}$C overnight. Finally, the samples were calcined. A schematic representation of the synthesis can be found in the Appendix A (see Appendix A).

The quantities of precursors used in the synthesis with 1 eq of citric acid in a 250 mL autoclave were: 6 mmol of citric acid, 6 mmol of Bi(NO${}_{3}$)${}_{3}$(H${}_{2}$O)${}_{5}$, and 6 mmol Fe(NO${}_{3}$)${}_{3}$(H${}_{2}$O)${}_{9}$ dissolved in a 120 mL solution of ethylene glycol and water and 0.54 g of urea. The amounts were adjusted accordingly for differently sized autoclaves.

### 3.2. Characterization Techniques and Equipment

The structure and phase purity of the synthesized materials were characterized using a Bruker D2 Phaser X-ray diffractometer with a 1.54184*Å* copper tube. Using the DIFRACC.EVA V4.3.1.2 software, a semi-quantitative analysis of the diffraction pattern was performed to identify secondary phases. The morphological analysis of the sample was performed using scanning electron microscopy and energy-dispersive X-ray spectroscopy. A field emission electron microscope MIRA 3, TESCAN equipped with a Bruker X-Flash 6–30 detector with a resolution of 123 eV in Mn Kα was used. The diffuse reflectance spectrum was measured by UV-Vis spectroscopy (Perkin Elmer, Waltham, MA, USA) with λ∈ 200–1000 nm with an integrating sphere. These spectra were transformed by a Kubelka–Munk model in order to estimate the band gap value.

### 3.3. Photocatalytic Experiments

We measured the photocatalytic activity of the samples at room temperature using Rhodamine B as the model dye with an initial concentration of 5 mg/L at pH = 7. In a typical experiment, 50 mL of dye solution was mixed with 50 mg of the BiFeO${}_{3}$ sample. The reaction mixture was stirred in darkness for 60 min to ensure an adsorption–desorption equilibrium between the catalyst and the dye solution. Then, the samples were irradiated with four Kessil lamps with an emission wavelength of λ=427 nm (2x PR160-427 nm) and λ=440 nm (2x PR160-440 nm) with a distance of exactly 10 cm with respect to the center of the reaction. After every 30 min, the catalysts were separated from the reaction mixture by means of centrifugation (1000 rpm for 3 min). The dye concentration as a function of time was determined based on the Lambert–Beer equation by measuring the absorbance of light at the maximum intensity of the absorption peak.

For the reuseability study the catalyst was isolated by means of centrifugation, washed repeatedly with water/ethanol, and then dried overnight at 80 ${}^{{}^{\circ}}$C. The catalyst was then reused in four consecutive photocatalysis experiments following the procedure mentioned above.

The absorption spectrum of Rhodamine B was measured using a UV–Vis spectrophotometer GENESYS 30TM with tungsten-halogen light source and silicon photo-diode detector. The spectra were fitted with the Thermo Scientific VISIONlite PC software suite.

## 4. Conclusions

In summary, for the first time, bismuth ferrite hollow spheres were synthesized by a facile hydrothermal method. X-ray diffraction studies show that as the time and temperature of the hydrothermal process increase, additional phases such as Bi${}_{2}$Fe${}_{4}$O${}_{9}$ and Bi${}_{2}$O${}_{3}$ form, and the overall quantities of secondary phases increase. Additionally, with an increase in treatment time, crystals that form the spheres become larger, while the overall porosity and the diameter of the spheres increase. Furthermore, the spheres become more irregularly shaped and tend to grow into each other forming larger microstructures. The addition of 1 equivalent of citric acid is sufficient for the formation of highly crystalline BiFeO${}_{3}$ hollow spheres, whereas additional citric acid lead to an increase in Bi${}_{2}$Fe${}_{4}$O${}_{9}$. By determining and applying the correct calcination pathway, we were able to isolate BiFeO${}_{3}$ hollow spheres, which are characterized by a rhombohedral perovskite phase with R3c symmetry. Unwanted secondary phases could be removed by washing the samples with acetic acid and nitric acid without affecting the hollow sphere morphology. For optimal reaction conditions, we observed BiFeO${}_{3}$ hollow spheres with a diameter of approximately 200 nm to 1.5 μm and a wall thickness of 50 nm. Diffuse reflectance UV–vis spectra were used to analyze the optical properties of the hollow spheres. The synthesized BiFeO${}_{3}$ hollow spheres absorb a large amount of light in the visible-light region and are, furthermore, characterized by a band gap of 2.22 eV. The spheres were used in photocatalytic degradation experiments using Rhodamine B as the model organic pollutant under visible-light irradiation. The BiFeO${}_{3}$ hollow spheres are characterized by an exceptionally high photocatalytic activity leading to complete dye degradation in less than 150 min. To the best of our knowledge, the catalytic activity of the unsupported BiFeO${}_{3}$ hollow spheres in the present system is unmatched in current literature. Faster complete degradation was achieved by slightly increasing the catalyst concentration. Furthermore, the catalyst recycle experiment shows that the semiconductor spheres are stable under the applied reaction conditions and can be recycled successfully without loss of photocatalytic performance. The enhanced photodegradation activity of the BiFeO${}_{3}$ catalysts can be attributed to their hollow sphere morphology. Additionally, by trapping experiments we show that photogenerated holes (h+), hydroxyl radicals (• OH), and superoxide radicals (• O${}_{2}^{-}$) are the main active species in the photodegradation process.

## Figures and Tables

**Figure 1 molecules-28-05079-f001:**
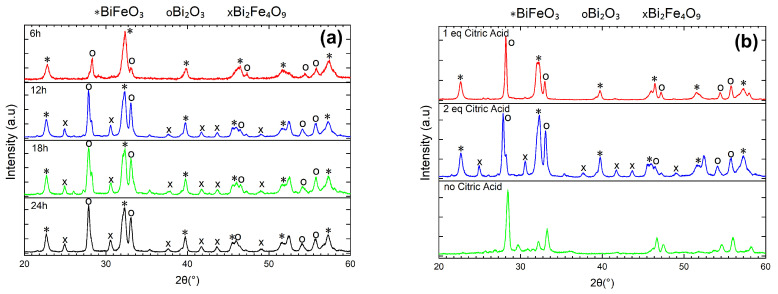
X-ray diffractograms of BiFeO${}_{3}$ samples resulting from (**a**) different hydrothermal treatment times (**b**) different metal/citric acid ratios.

**Figure 2 molecules-28-05079-f002:**
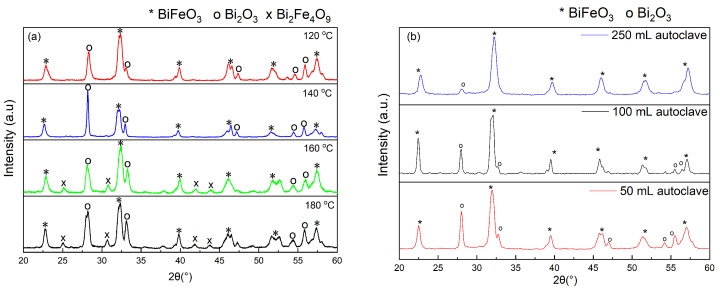
X-ray diffractograms of BiFeO${}_{3}$ samples resulting from different (**a**) hydrothermal treatment temperatures and (**b**) autoclave sizes (pressures).

**Figure 3 molecules-28-05079-f003:**
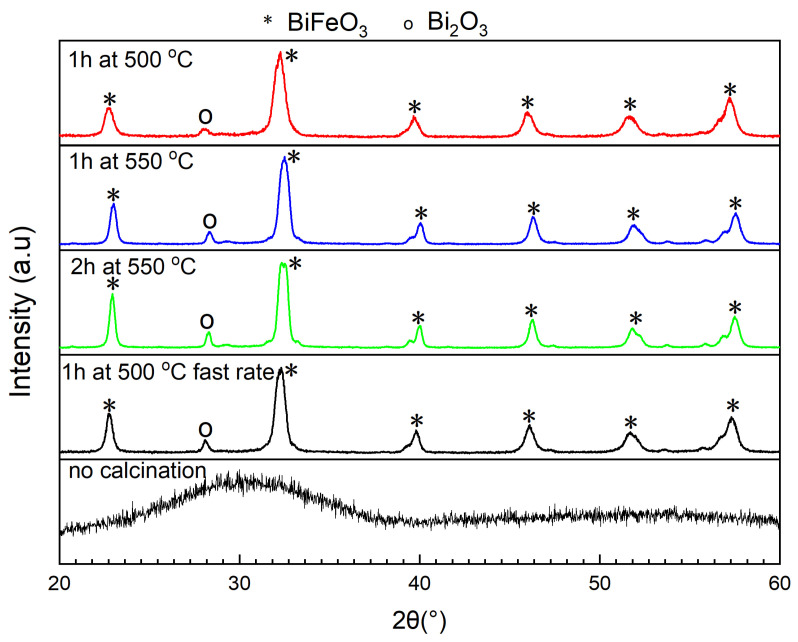
X-ray diffractograms of BiFeO${}_{3}$ samples varying the calcination path.

**Figure 4 molecules-28-05079-f004:**
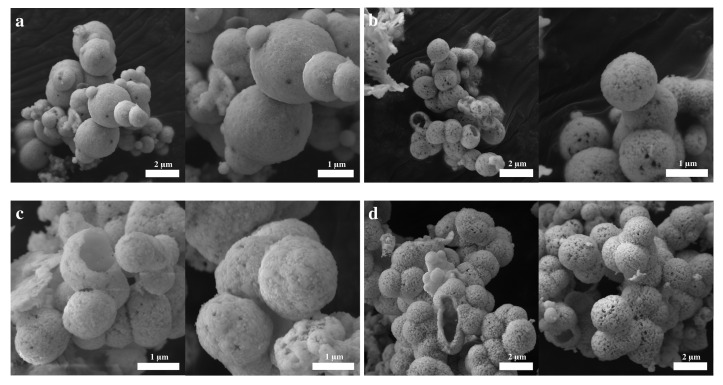
SEM images of BiFeO${}_{3}$ samples with different hydrothermal treatment times: (**a**) 6 h; (**b**) 12 h; (**c**) 18 h; (**d**) 24 h.

**Figure 5 molecules-28-05079-f005:**
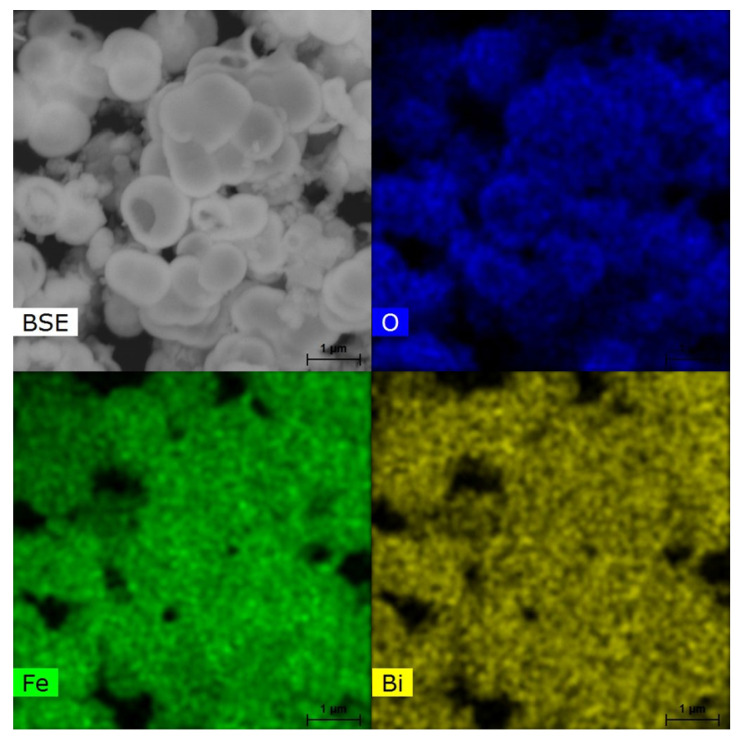
Elemental EDS mapping of BiFeO${}_{3}$ hollow spheres.

**Figure 6 molecules-28-05079-f006:**
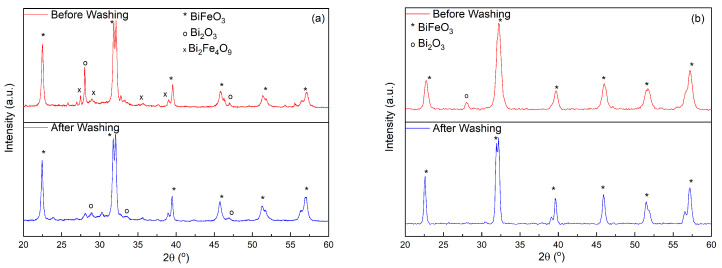
X-ray diffractograms of BiFeO${}_{3}$ samples before and after washing with acetic acid and nitric acid (**a**) and acetic acid (**b**).

**Figure 7 molecules-28-05079-f007:**
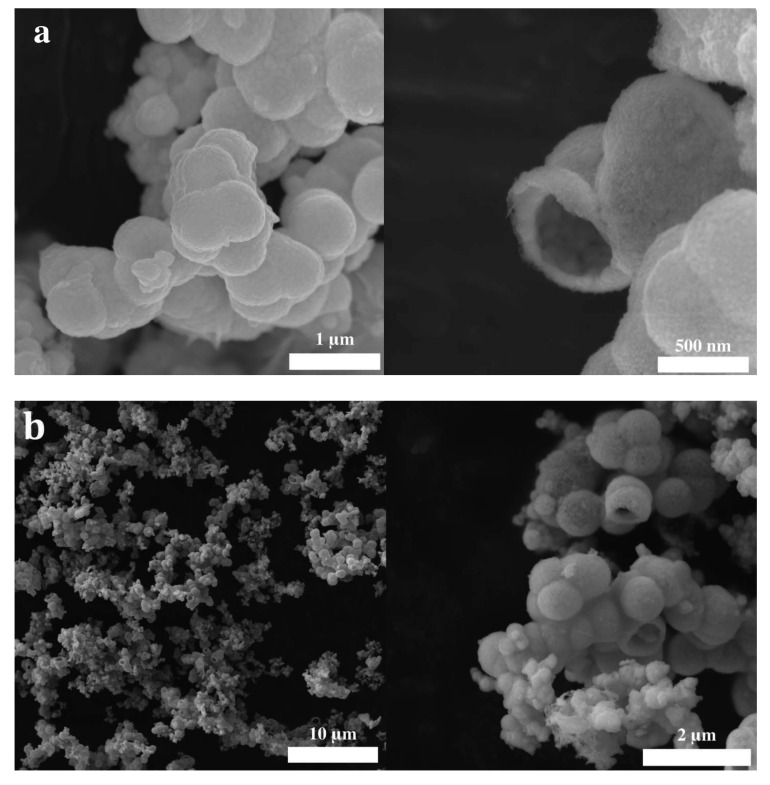
SEM images of BiFeO${}_{3}$ samples (**a**) before and (**b**) after the washing with acetic acid and nitric acid.

**Figure 8 molecules-28-05079-f008:**
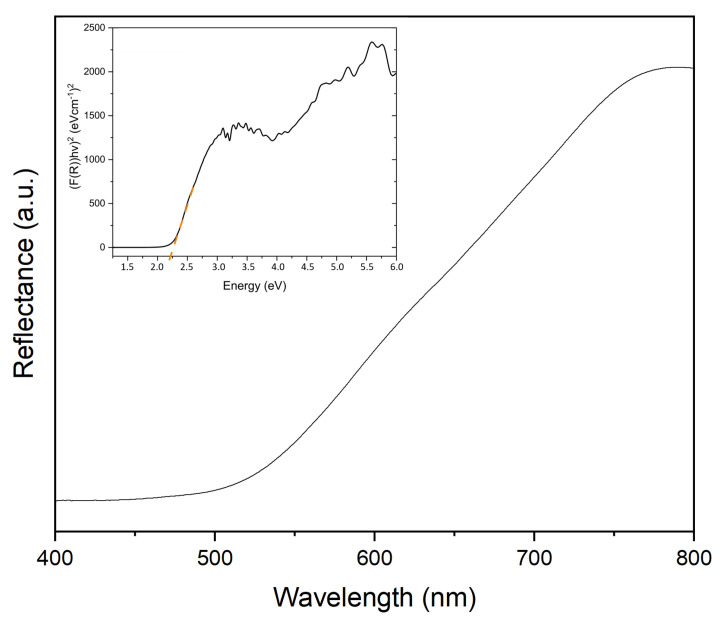
Absorbance spectra and Tauc plot (inset) for BiFeO${}_{3}$ hollow spheres.

**Figure 9 molecules-28-05079-f009:**
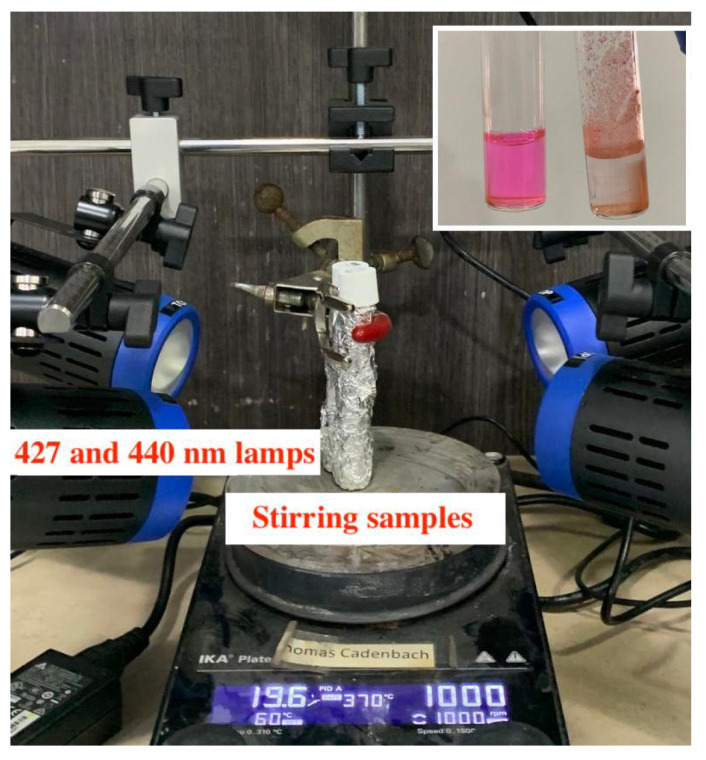
Setup of a photocatalysis experiment.

**Figure 10 molecules-28-05079-f010:**
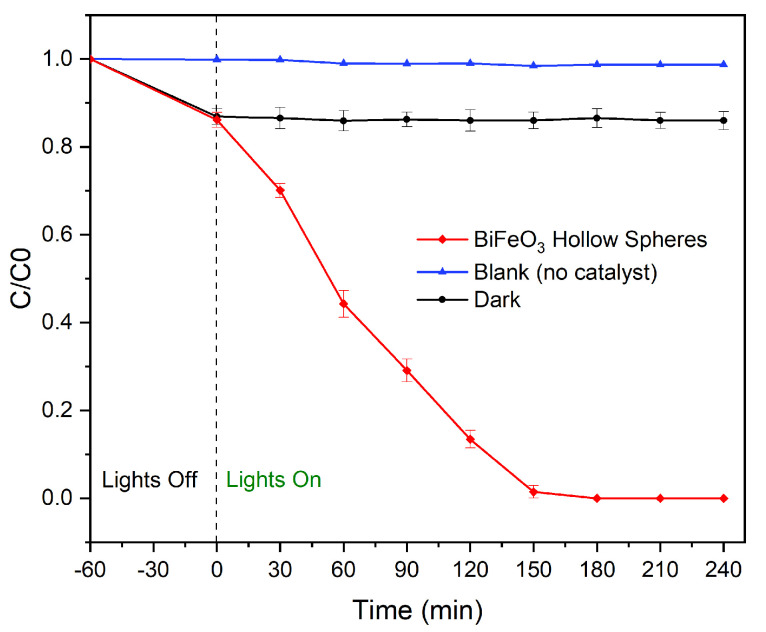
Removal of Rhodamine B as a function of irradiation time under visible-light using BiFeO${}_{3}$ hollow spheres.

**Figure 11 molecules-28-05079-f011:**
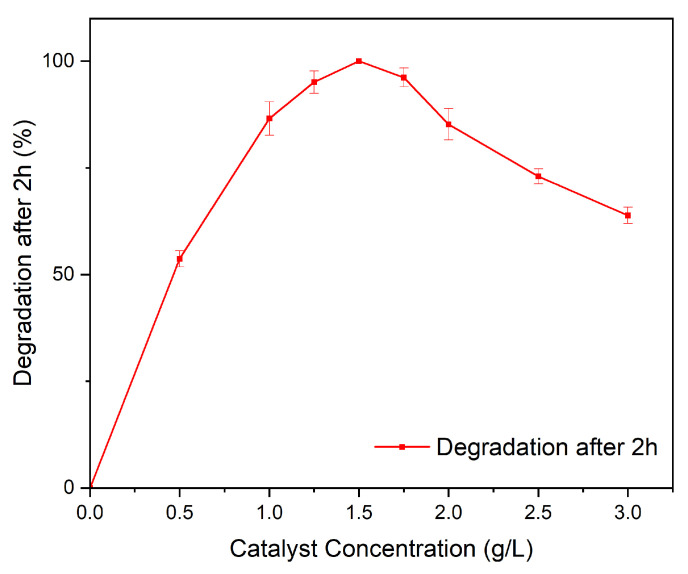
Effect of catalyst concentration on the degradation of Rhodamine B.

**Figure 12 molecules-28-05079-f012:**
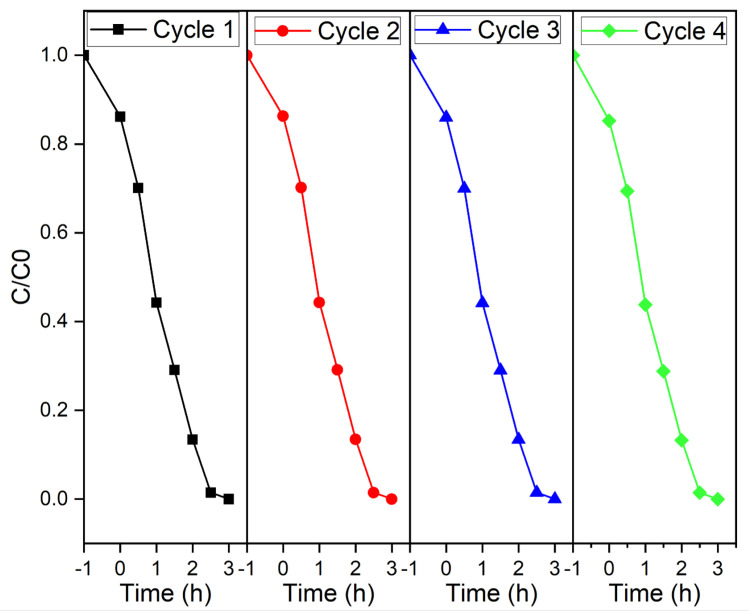
Removal of Rhodamine B using BiFeO${}_{3}$ hollow spheres in four consecutive cycles.

**Figure 13 molecules-28-05079-f013:**
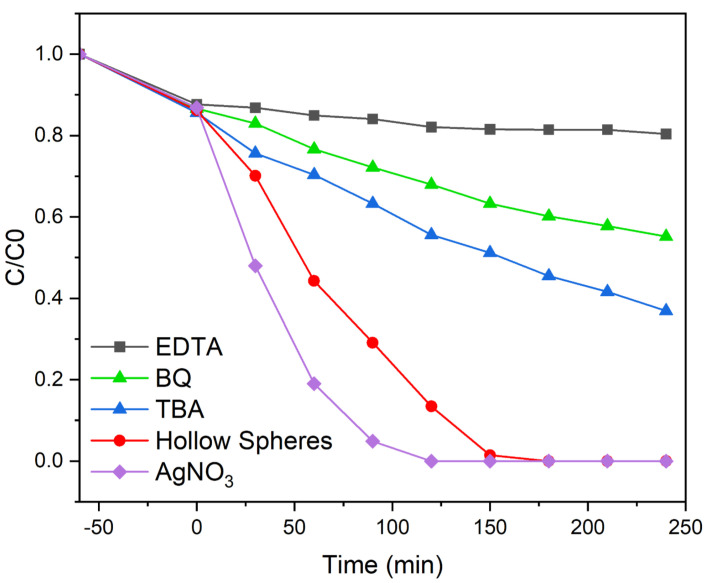
Trapping experiments for the removal of Rhodamine B using BiFeO${}_{3}$ hollow spheres.

## Data Availability

Raw data available on request.

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
