# Peer review of "Hydrothermal Synthesis of Bismuth Ferrite Hollow Spheres with Enhanced Visible-Light Photocatalytic Activity"

_molecules, 2023, doi:10.3390/molecules28135079_

Round 1

Reviewer 1 Report

Photo responsive materials exhibited great potential in photovoltaic cell, CO2 reduction, hydrogen releasing, light therapy, and pollutants removing. Bismuth ferrite attracted great attention due to its relative narrow band gap. In this manuscript, the authors reported the synthesis of BiFeO3 hollow spheres by a facile citric acid/ urea/ethylene glycol assisted hydrothermal method. They characterized the synthesized materials using SEM, XRD and UV-DRS, and found that they exhibited excellent photocatalytic performances in Rhodamine B degradation. The synthetic method is interesting and the result is accepted, whereas more detailed evidences should be provided. So I recommended it to be published in Molecules after a major revision.

1.       How about the reproducibility of the materials since the purity and morphology of the synthesized materials are very sensitive to the synthesis parameters?

2.       TEM and HRTEM should be applied to show the hollow structure and arrangement of different crystals including BiFeO3, Bi2O3 and others.

3.       Where does the reaction take place? Surface or the core? BET and BJH measurements should be provided to determine the surface area and pore size.

4.       Error bars should be provided in the point plots, such as Figure 10 and 11.

5.       To better understand the photocatalytic mechanism, direct characterizations of the active species including ⸳OH and ⸳O2- should be provided.

Author Response

We would like to thank the Reviewer for the fruitful comments and suggestions. Please see the attachment for a point-by-point response to the comments.

Reviewer 2 Report

The work by Cadenbach et. al. reported the preparation and photocatalytic property of BiFeO3 hollow spheres. The following issues should be addressed:

1) Error bars should be provided in the figures.

2)The initial concentration of RhB should be provided.

3)The EDS mapping result was shown in Figure 5, but the EDS spectrum was missing. Please provide this spectrum to support the argument “Bi, Fe, and O with an approximate ratio of 1:1:3”.

4) Only two autoclave sizes were used when discussing the effect of pressure, and impurity still existed when using the size of 250 mL. Is it possible to use a larger-sized autoclave?

5) The photocatalytic property of BiFeO3 after washing with acid should be investigated.

6) Several format and typographic errors can be found. For instance:

a) Abstract: “in the range of 200 nm to 1.5 µm and a wall thickness of 50 nm”. Inconsistent with the data in the main text.

b) Line 246, missing “-” between the two wavelengths.

c) Line 265, “C/C0,”

d) Line 334, wrong sign for the superoxide radical.

Author Response

(The authors gave the same response as above.)

Reviewer 3 Report

In the present manuscript, the authors demonstrated the synthesis of BiFeO3 hollow spheres by a facile hydrothermal and the mechanism of formation of pure phase BiFeO3 hollow spheres is investigated systematically by variation of synthetic parameters. To evaluate the photocatalytic efficiency, experiments were carried out on the decolourization of aqueous solutions of the RhB dye. The present study provides some valuable information and the content is very significant in this field. However, the manuscript need a systematic modification and I recommended a major revision of the article from its present form before it can be published. Some specific comments are as follows:

1. The abstract and conclusion sections should be a specific and scientific approach.

2. The authors should explain the limitation and novelty of this investigation have to be clarified in more detail.

3. The motivation part is a lack of introduction. The authors should revise the introduction part.

4. The authors should provide the schematic representation of experimental methods.

5. What is the pH of the reaction solution? The pH of the solution normally varies from precursor to precursor. The authors must justify the selection of pH, temperature and time.

6. In the XRD, it is better to mention the JCPDS card numbers.

7. Authors are advised to include the atomic and weight% values.

8. Authors should perform XPS analysis.

9. What is the key factor affecting efficiency?

10. Authors should mention the parameters of light source.

11. Authors should explain the photocatalytic mechanism with a clear schematic representation.

12. How do these results influence the previously reported results?

13. Grammar and spell errors existed in the manuscript. Therefore, the authors are advised to recheck the whole manuscript for improving the language and structure carefully.

Grammar and spell errors existed in the manuscript. Therefore, the authors are advised to recheck the whole manuscript for improving the language and structure carefully.

Author Response

(The authors gave the same response as above.)

Round 2

Reviewer 1 Report

It can be accepted at the present form.

Author Response

We would like to thank the Reviewer again for his fruitful and professional review process. We are glad that he recommends the publication of the article in its present form. 

Reviewer 2 Report

The reviewer has addressed the issues raised by the reviewer previously. I suggest the publication of this manuscript in its current form.

Author Response

(The authors gave the same response as above.)

Reviewer 3 Report

In the revised version, the manuscript has been greatly improved. However, the manuscript still needs major changes.

1. The authors should produce the DRS analysis for all samples.

2. The authors should produe the PL spectra of all samples.

3. The authors should calculate the valence band and conduction band values.

4. The authors should present a proper photocatalytic mechanism with neat schematic representation.

5. The authors should include the atomic and weight% values in the EDS spectrum.

6. The authors should produce a schematic representation for the synthesis mechnaism.

7. The authors should check the labels in figures and abbrevations.

8. The manuscript still needs a proffisional english service. 

The manuscript still needs a proffisional english service. 

Author Response

We would like to thank the reviewer for his comment that the manuscript has been greatly improved. With this new revision, we are positive that the article now meets the standards of Molecules.

Please see the attachment for our point-by-point response.
